# Two Waves of Specific B Cell Memory Immunoreconstruction Observed in Anti-HHV1–3 IgG Kinetics after Hematopoietic Stem Cell Transplantation

**DOI:** 10.3390/biomedicines12030566

**Published:** 2024-03-03

**Authors:** Przemyslaw Zdziarski, Andrzej Gamian

**Affiliations:** 1Lower Silesian Center for Cellular Transplantation, 53-439 Wroclaw, Poland; 2Clinical Research Center PRION, 50-385 Wroclaw, Poland; 3Hirszfeld Institute of Immunology and Experimental Therapy, Polish Academy of Sciences, 53-114 Wroclaw, Poland; andrzej.gamian@hirszfeld.pl

**Keywords:** virome, human herpesvirus, varicella zoster virus, herpes simplex virus, Tzanck test, IgG protective level, pathogenesis, immunodeficiency, hematopoietic stem cell transplantation, hematopoietic cell transplant comorbidity index, statistical bias

## Abstract

Background: Humoral memory and specific antibody levels depend on the kind of antigen and individual immunofactors. The presence of IgM antibodies or a fourfold rise in specific IgG levels are generally accepted as diagnostic factors in the serology of acute viral infections. This basic model is not adequate for the herpes virome, especially after hematopoietic stem cell transplantation (HSCT), due to continuous, usually multifocal antigenic stimulation, various donor serostatuses, immunosuppression, and individual immunoreconstitution. Methods: A case–control study was conducted to identify active infection cases of human herpesvirus (HHV) (from 300 diagnosed immunocompromised patients) and to evaluate historically associated humoral factors to look at outcomes. We considered only the data of patients with meticulous differential diagnosis to exclude other causes, and thereby to observe pathways and temporal relationships, not the statistical ones usually collected in cohorts. Despite the small number, such data collection and analysis methods avoid a number of biases and indicate cause and effect. Results: In this observational study, a retrospective analysis of data from 300 patients with clinical diagnosis of herpes simplex virus (HSV) and varicella zoster virus (VZV) reactivation showed a number of biases. Two well-differentiated cases (confirmed by a Tzanck test) with various diseases and conditioning evolutions of immune parameters showed an interesting pathway. Exponential decreases in specific IgGs after HSCT preceded virus replication were observed, with a cytopathic effect (shingles, VZV encephalitis and HSV-induced mucositis). The minima (lowest IgG levels) before herpesvirus reactivation were 234.23 mIU/mL and 94 RU/mL for VZV and HSV, respectively. This coincided with a low CD4 titer, but without other infectious processes. Other immune response parameters such as Treg, cytotoxic T cells, and complement and total IgG level were the same as they were before the transplant procedure. Interestingly, a second wave of immunoreconstitution with an anamnestic antibody response was not always observed. It coincided with prolonged herpes viral infection. A patient with lymphocyte depletion in conditioning showed an earlier second wave of immunoreconstitution (6th vs. 14th month). Conclusions: As is typical for infancy, the kinetics of the IgG level is unique after HSCT (the decline phase is first). Host microbiome factors (e.g., HHV1–3-serostatus) should be taken into account to predict risk of non-relapse mortality and survival after HSCT. The levels of specific antibodies help in predicting prognoses and improve disease management. A lack of differentiation and the confusing bias of the assessor (i.e., observer selection bias) are the main obstacles in statistical HHV1–3 research. Such time-lapse case studies may be the first to build evidence of a pathway and an association between immune parameters and HHV disease.

## 1. Introduction

Human herpesvirus (HHV) infections are widespread diseases that are benign in immunocompetent people but may lead to severe complications and occasionally to death in severely immunocompromised people. A decline in maternal anti-VZV antibodies in infants with chickenpox (the cut-off value was estimated to be 150 mIU/mL) has already been described [1]. Further observations showed a significant inverse correlation between varicella severity and the level of maternal anti-VZV IgG [2]. The same correlation was observed for neonatal herpes simplex virus (HSV) [3]. Analogous conclusions are being drawn from epidemiology, e.g., the discrepancy between a high shedding rate among pregnant women with established HSV infection and a low neonatal transmission rate [4]. Successful prevention as a result of the passive transfer of maternal antibodies (class IgG only) indicates that the humoral response in HHV1–3 infection is pivotal. This simple model indicates the threshold of protection against primary infection, but for HHV reactivation, the protective level has not been estimated so far; insufficient data exist on the relationship between viral reactivations and immune reconstitution [5]. Furthermore, concentrations of specific IgGs have been used to estimate immunity against varicella zoster virus (VZV) in solid organ transplant (SOT) recipients only [6]. However, in hematopoietic stem cell transplantation (HSCT), the situation is more problematic: HHV1–3 serostatus is not evaluated and acyclovir is overused [7]. The relationship between preexisting anti-VZV levels and clinical VZV reactivation in HSCT recipients was studied in a small group of 13 subjects without a significant difference found between sick and healthy subjects; however, VZV reactivation was arbitrarily diagnosed with a non-validated, semi-quantitative immune adherence hemagglutination method. Differentiation and anti-HSV1/2 testing were not performed. Herpes and limited ocular zoster were indistinguishable. It is of note that the small group was also non-homogenous in terms of transplant procedures (they received auto- or allo-HSCT); therefore, significant biases were present during data collection [8]. A decrease in all immunoglobulin subclasses was observed in patients less than 100 days after HSCT and normalized after a year. Patients with low IgG levels showed a decreased survival rate and increased incidence of transplant-related mortality despite low HSCT comorbidity index (HCT-CI) scores [9]. The HCT-CI was shown to predict the risk of no-relapse mortality (NRM) without the analysis of virome risk factors (e.g., HHV1–3 serostatus). Furthermore, most studies have focused on cytomegalovirus (CMV); however, in these studies, there is no well-defined cytomegalovirus disease, which is diagnosed based on molecular tests (usually high viral load), contrary to the development of the disease process (pathogenesis) [10]. Such biases make prospective cohort studies difficult in the absence of a clear case definition and patient selection, which are critical for effective investigation. Multifactorial area case studies are a good tool for exploring rare diseases or when other study types are focused on a few of the most likely causative factors (in this case, HCT-CI) [11]. In a cohort study, a statistical association between a risk factor and a disease or condition is studied, as opposed to studying to the causes of the disease or condition. This is because it may be difficult to determine the timing of the risk factor relative to the onset of the disease. Case studies make it possible to look at multiple risk factors at once (timing) [10,11]. This is a completely different approach to regular research and data analysis, enabling translational medicine and converting discoveries all the way from bench to bedside.

The aim of this study is to characterize the pathway and pathogenetic relationship between the anti-HHV IgG level and the clinical presentation (disease chronicity and severity) in patients with herpesvirus diseases, i.e., the shift from latency to the replication of HSV and VZV (i.e., HHV1/2 and 3, respectively).

## 2. Materials and Methods

### 2.1. Material

The observational study involved gathering the data histories of 300 patients with HHV1–3 reactivation. Only patients diagnosed with humoral immunodeficiency with well-documented previous HSV or VZV primary infections, as reported in interviews, were included. The patients comprised a heterogeneous group, experiencing various primary or secondary immunodeficiencies (detailed in the Appendix A). Simple statistical analysis could not be used due to a number of biases [8]. Firstly, people were not randomized into their outcome groups; secondly, they were dissimilar in their demographic and clinical characteristics. Therefore, the study was prone to confounding data [12].

Secondly, the symptoms of herpes zoster and eczema herpetiformis often overlap with graft-versus-host disease, autoimmunity, drug allergies, and other infectious complications in immunocompromised patients (HHV1–3 reactivation was a diagnosis of exclusion). Thus, the data from patients with mis-differentiation, without Tzanck test [13] for confirmation and excluding other causes, were not included in the study. Consequently, most of the patients were disqualified (Appendix A).

### 2.2. Time-Lapse Data Collection

Our previous study adopted Pinquier’s model [1] for CMV reactivation during infancy and introduced it during cytomegalovirus reactivation in patients who have undergone transplant after HSCT [14]. The current study investigated the role of herpes simplex 1/2 and varicella-zoster-specific IgG (i.e., HSV and VZV, respectively) The flow chart of the systematic clinical observation and data collection before and after transplantation is presented in Figure 1.

### 2.3. Methods

In our center, serum samples were collected during every clinical visit. Blood samples (0.5 mL) were collected in a dry tube and centrifuged for 10 to 15 min at 3000 rpm. After centrifugation, the serum was extracted and stored at −20 °C. The specific anti-HHV antibodies were tested with EUROIMMUN ELISA kits (Lübeck, Germany). Recombinant glycoprotein C1(GC) HHV1 and virus of the “VZ-10” strain (highly purified cell lysate from MCR-5 cells infected with varicella zoster) were used as an antigen. The data were expressed in accordance with the international reference preparation W1044 (WHO) as IU/L (i.e., mIU/mL), as described by previously [1,2]. Neither of the reference standards were completely defined by the manufacturer for HSV [4]; therefore, the data are expressed in RU/mL, as described by the manufacturer. The positive results are >110 IU/L and >22 RU/mL, and the negative results are <80 IU/L and <16 IU/mL, for anti-VZV and anti-HSV IgG, respectively. According to the manufacturer’s description, the results between these values were considered to be equivocal. The affinity IgG was presented as a relative avidity index (RAI), as described by the manufacturer, and expressed in %.

Leukocyte count analyses were performed by the Sysmex Automated Hematology System (SYSMEX, Kobe, JPN). Flow cytometry was performed using a FACS Calibur flow cytometer (Becton Dickinson, San Jose, CA, USA). According to the standard operating procedure in our center, after the CD45-based gating of the lymphocytes, the lymphocyte subsets were calculated by multiplying the percentage of these lymphocyte subpopulations by the lymphocyte counts [14].

The patients monitored in our outpatient clinic were systematically serologically tested for their IgG levels against HHVs, i.e., CMV, EBV, VZV, and HSV. The cut-off value was estimated when the patients developed reactivation (Figure 1). 

## 3. Results

In the preliminary study, our analysis of the medical history of 300 patients diagnosed with HSV or VZV reactivation showed a number of biases [12]. In most patients with HSV and VZV, the diagnosis was not confirmed, and other diseases were not excluded (see Section 2.1 and the Appendix A). Therefore, ascertainment and confusing bias were the most common approaches in analyses and data collection methods [12]. Multicenter cohort studies have the same disadvantages, as each researcher makes a diagnosis in a unique way without differential diagnosis and ruling out other causes, e.g., drug allergy; graft-versus-host disease (GVHD) (see below).

If the individual fates of patients are analyzed, then exposure to various known factors (e.g., HHV1–3) and individual outcomes give the opportunity for obtaining additional data, and each case becomes a “nature experiment”. This brings our observation closer to experimental research if the number of potential factors (usually overlapping) in differential diagnosis is significantly reduced.

Finally, two HSCT cases with well-documented previous (primary) infection who later underwent HHV reactivation with a cytopathic effect, i.e., positive Tzanck test (presence of acantholytic and multinuclear giant cells), were considered [13]. In this way, the molecular phenomena and the HHV1–3 infection were documented, as were the pathological process, i.e., the disease development with cytopathic effect. Time-lapse serological data collection is presented in Figure 1.

Clinical immune parameters and HSCT data from the two cases are summarized in Table 1.

Patient 1 was admitted to our center, presenting with a severe herpesvirus complication. Earlier, the patient received hematopoietic stem cell transplantation from a matched, unrelated donor (10/10) with standard reduced intensity conditioning (RIC).

After full chimerism (i.e., +28 day), laboratory tests revealed the donor blood group without hemolytic anemia and the patient was in a good condition of health. The patient was discharged from hospital with follow-up visit every two months. GvHD prophylaxis consisted of cyclosporine A (CsA) with a stable level and good efficacy (Table 1). Therefore, the patient did not receive prednisone for GVHD. One year after the transplant, the patient developed zoster (facial, then diffuse), complicated with viral encephalitis despite the immediate administration of a high dose of acyclovir. Local skin and mucous membrane blisters also developed with stage 2–4 oral mucositis, predominantly in the mouth and nose area.

Severe CD4 T and B lymphopenia was observed below 100 and 20 cells/μL, respectively. However, other signs of abnormal CD8+ cytotoxic T cell level and Tc-mediated immunity were not observed. The VZV and HSV reactivation coincided with low CD4 Th and CD20 B cells titer (Table 1), but the CMV and EBV PCR results were negative. Retrospective analysis of serum samples showed specific anti-HSV and anti-VZV IgG level fluctuation. Total IgG level was stable (about 1000 mg/dL) (Figure 2).

Exponential kinetics of specific anti-VZV IgG level (exponential decrease and increase within eighteen-month observation) was presented with parabolic trend-line. Contrary to anti-VZV, prolonged herpes mucositis was observed with a significant (also exponential) decrease in anti-HSV. Despite a temporary increase in the first months, the trend line of anti-HSV was inverted hyperbola (exponential function type a^x^, where a < 1) that coincided with the late onset and the increase in the severity of the prolonged mucositis (Figure 2). Contrary to the exponential change of HHV-specific IgG, the total IgG level was stable, without significant extrema. First- and second-wave immunoreconstitution was observed only for VZV-specific IgG, contrary to anti-HSV (only first) and total IgG (no significant peak value).

The lowest IgG level before HHV reactivation was 234.23 mIU/mL (IU/L) and 94 RU/mL for VZV and HSV, respectively. However, the total IgG level was within reference range, but the serum level of -β2-microglubulin was high (Table 1). Host-VZV dynamics showed acute and resolving infection, i.e., setting where VZV initiated acute infection that then rapidly demonstrated very high anti-VZV IgG level (till 4515 IU/L) with shift to the latency. The low and decreased levels of anti-HSV IgG corresponded with prolonged oral mucositis (II → IV⁰). No complement abnormality was observed, nor was any sign of GvHD at +100 day or later (Table 1). As the observed entire abnormality of the skin was unequivocally explained by a non-GVHD documented cause, the skin eruption was not included in the calculation of the global severity [9,15]. Contrary to VZV-specific antibodies, the level of vaccine-induced anti-HBs was high, and total IgG levels were stable (Table 2).

Patient 2 developed severe oral mucositis and diffuse zoster after HSCT; these occurred in the second and third months after HSCT, respectively. The patient’s initial serological analysis showed high HHV-specific IgG levels before HSCT. Under the effect of immunosuppressive therapy (with ATG—Table 1), the coincidence of VZV reactivation, progressive oral mucositis, and the decrease in the levels of HSV- and VZV-specific IgGs was observed earlier. The inverse correlations between the severity and serological parameters were found during progression as well as oral/cutaneous healing (U-shaped parabolas, Figure 3). Total IgG levels were not stable (between 300 and 1000 mg/dL), and had significant peak values (extrema).

The first wave of immunoreconstitution showed short-term immunoprotection. The second wave gave significant (also exponential and more than 10-fold) and specific increases in antibodies with long-term protection capabilities. Contrary to Patient 1’s experience, the second wave of immunoreconstitution was observed earlier (about 6 months after HSCT) and for both the antibodies (HSV- and VZV-specific antibodies); however, the less intensive second wave corresponds with the prolonged zoster (Figure 3). The increase in anti-HSV IgGs preceded oral mucosal barrier healing (after 7 months). The first and second waves (to a lesser extent and not exponentially) were observed in the extrema of the total IgG level.

## 4. Discussion

### 4.1. Technical Aspect and Clinical Model

In HSCT, contrary to primary infectious process, when the virus replication is local (in portal of entry), the HHV reactivation is systemic [14]. The exposure type is important; whether it is local or systemic exposure may be important, as our cut-off level for VZV reactivation is higher than that previously described [1,2]. The subtle difference is crucial when we look at immunoprophylactic (vaccination) or immunotherapeutic regimens.

Secondly, the methods of analysis and cut-off titer definition are different. Commercial assays of antibodies against HSV are not validated [4]. Furthermore, the discrepancies between the protective level (i.e., 150 IU/L) [1,2] and the cut-off value, which are reported to facilitate testing by manufacturers (e.g., 100 IU/L for Euroimmun), indicate that determining the cut-off values for the presence of disease vs. the absence of disease is performed empirically in laboratory practice. It is not a protective level. Protection after vaccination of a specific patient (e.g., after HSCT) is more difficult to achieve than population immunity. This is a key question which it will change the philosophy of immunoprophylaxis (social or individual goals) and the definition of the “protective level” (see below).

Thirdly, two images from the same patient might look very different [16]. Since molecular testing is rarely used for HHV1–3 infections, even in immunocompromised patients, availability is limited [14]. Here, VZVHHV1–3 reactivation was firstly used in the diagnosis of exclusion, secondly, it was indicated by the acantholytic and multinucleated giant cells in the image (Tzanck test) [13,16]. Due to the rapid turnaround time and correlation with clinical symptoms, serologic methods are preferable for VZV1-3 in contrast to viral isolation [17]. Furthermore, in infectious disease, a positive specific IgG result is a well-established biomarker of recovery, without a quantitative relationship (a fourfold rise in specific IgG level is generally accepted as diagnostic in serology) [15]. A frequently used technique in identifying the direction of effects involves utilizing longitudinal data in which laboratory parameters have a temporal order to them. Our time-lapse observations and immunomonitoring data for patients who have undergone transplants are valuable in assessing the immune status (immunoreconstitution) and prediction of disease.

### 4.2. Immunomodulation and Host–Virus Interaction in Patients Who Have Undergone Transplants

Herpesviruses, such as HHV1–3, through the use of different genetic programs (latency versus lytic replication), are conditionally cytopathic and only cause tissue damage upon entry into the lytic cycle. For latent HHV1–3 replication, immunological ignorance is the first strategy in avoiding immune responses (HSV and VZV preferentially infect immune-privileged sites). Furthermore, a prospective study of 281 allogenic HSCT recipients found a 3-year cumulative incidence of 6.3% for HHV–encephalitis due to herpesvirus infections [18].

It is noteworthy that the β2-microglobulin (invariant chain of HLA class I) level was high and that the peripheral blood lymphocyte subset (percentages) showed normal or very high cytotoxic T cells (CD8+) (Table 1) as compared with donors without the illness [19]. Patients lacking β_2_M expression show humoral disorders with low IgG and severe skin diseases [20].

The humoral counterpart is critical, as presented previously for VZV [1,2] and HSV [4]. Major immunomarkers (e.g., CD4, total IgG level) did not reflect a specific response formation (Table 1). Although the transplantation procedure itself (conditioning, immunosuppression) is standardized, the course and immunological reconstitution in Patient 1 was far from the general scheme, since anti-HSV was not seen in the second wave of immunoreconstruction (Figure 2). The vigorous immune response to VZV (a virus close to HSV) was not protective against HSV or severe mucositis, in spite of the cross-reactive antibodies between HSV-1 gB and VZV gp-II [21].

Interestingly, clinical symptoms were observed after a rapid decrease in specific IgG. In our patients, the increase in specific IgG and its kinetics shows progress from a primary immune state to the memory state, as described elsewhere [22]. Therefore, observation of host IgG dynamics may be a hallmark of HHV’s shift from latency to replication, as well as being indicative of the preclinical initial stage of zoster or mucositis. Our model gives a tool for predictive and preemptive therapies. This is important because acyclovir is a nucleoside (guanosine) analogue which blocks HHV1–3 DNA synthesis. Furthermore, long-term use and abuse (e.g., depot form) leads to resistance. The costs of treating a developed disease with the cythopatic effect (illustrated by Tzank test) are much higher here, with reduced effectiveness. However, there is no vaccine against HSV, the efficacy of VZV immunization is not tested in a reproducible manner, and the effectiveness criteria vary [23]. The disease is diagnosed arbitrarily, and protection is assumed without laboratory analysis. Therefore, the vaccine’s effectiveness is studied with epidemiological and statistical methods (e.g., disease incidence and prevalence), contrary to the approaches of experimental studies. Our model may be the bridge between various philosophical approaches in medicine. However, even in the present socioeconomic context (population), HSCT and immunodeficiency (e.g., AIDS) involve very high costs, the treatment of infections and HHV1–3 complications (sometimes serious) have an impact on the NHS’s budget.

### 4.3. Host–Virus Dynamics

The classical primary antibody response proceeds in the following four phases after a foreign antigen challenge occurs [14,17]:Lag phase—very low/no antibody is detectableLog phase—the antibody titer increases logarithmically.Plateau phase—the antibody titer stabilizes.Decline phase—the antibody is catabolized.

Secondary (anamnestic) response exhibits the same four phases of primary response; however, after HSCT, this scenario, especially for the herpes virome, is not adequate (re-exposure vs. continuous exposure) [10,14].

Our observation shows two different known virus–host dynamics [24] occurring in parallel and simultaneously. The acute infection is characterized in a typical parabolic vertical trend line (Figure 3 and line for VZV in Figure 2), but chronic infection (long-term lytic replication) presents as an inverted hyperbola (line for anti-HSV in Figure 2). The delicate balance between the two host–virus dynamics is not visible in standard laboratory and clinical practice after HSCT. Interestingly, we observed two different virus–host dynamics of HHV1–3 (HSV1/2 and VZV, respectively) in one patient (P1) after HSCT; therefore, this can be the case under the effects of the same therapeutic regimen (Figure 2). In Patient 2 (contrary to Patient 1), VZV’s immunoreconstitution of HSV- and VZV-specific IgGs developed in a parallel manner (Figure 3). In HSCT, the presence of CMV-specific IgG (in donor) is a known, excellent predictive indicator of the likelihood of significant posttransplant infection [14]. There is no such observation to be made for HHV1–3. Our patients’ well-documented histories of herpes or chickenpox (strict selection presented in Figure 1 and the Appendix A) indicate the crucial role of anti-HHV1–3 IgG against reactivation (Figure 2 and Figure 3). Unfortunately, according to current practice, donors are screened for CMV only. Therefore, two different antibody kinetics of IgG against VZV and HSV in Patient 1 may indicate different donor serostatuses for VZV and HSV1/2, respectively.

The specific IgG that works against glycoprotein C of HHV1, as tested here, plays a crucial role in the infectious process, just in a different way. Glycoprotein C (GC) of HSV plays a principal role in the adsorption of the virus to the host’s cells [25] and blocks the complement cascade through C3b binding, inhibiting interactions with C5 and properdin and therefore inhibiting the formation of C5 convertase [26]. The passive transfer of specific IgGs against glycoprotein C did not protect complement-intact mice against HSV-1 [27]; here, the reduction in the titer was anti-GC-dependent [27]. In fact, the opposite phenomenon was observed in our patient: a decrease in the GC-specific antibody caused fast HSV replication. Oral mucositis was observed before the zoster (Figure 2). Therefore, the protective level for HSV-1 may be higher than that for VZV or other herpesviruses. A low concentration of anti-VZV or GC-specific IgG is required for opsonization (and ADCC), but in a higher titer; this leads to the blocking of adsorption and C3b binding by GC.

### 4.4. Humoral Compartment in Secondary Immunodeficiency after HSCT; Immunomodulation by Conditioning

The percentages of CD8 cells were normal, contrary to those of Th (CD4) and B (CD20) cells (Table 1). A good model for humoral immune reconstitution post-HSCT is the allogenic stem cell transplantation model, in the presence of common variable immunodeficiency (CVID). As shown in one study, the percentages of patients who withdrew from immunoglobulin replacement were 25, 40, and 60 percent in the first, second, and third years after HSCT, respectively [28]. Our observation coincides with the data from these previous studies. For our patients, a large decrease in specific immunoglobulin production also occurred between 6 months and 17 months (Figure 2 and Figure 3). Low immunoglobulin levels are associated with decreased survival, as described previously [29]. On the contrary, in P1, the total IgG level was stable (about 1000 mg/dL); only in P2 was two-wave humoral immunoreconstitution weakly observed in total IgG.

The explanation of the data may be the different times of reconstitution for several antigen-specific Th and B cells or the secretion of IgG by several plasma cells (Table 2). A second issue was ATG use in Patient 2. ATG directly affects not only T cells (especially Th with low CD4/8), but also B lymphocytes, so humoral immunoreconstruction occurs in a synchronous way; here, two maxima of the total IgG level are observed (Figure 3). The MAC regimen and cytostatic cause the inhibition of cell cycle arrest at different stages; therefore, lymphocyte renewal, with the selection of individual clones, resumes differently, especially in the second wave (Figure 2), with the various profiles of memory B cells.

#### 4.4.1. Memory B Cells 

Memory B cells (MBCs) are generated during the primary immune response. MBCs do not produce antibodies, unless re-exposure to antigen drives their differentiation into antibody-producing plasma cells. Interestingly, our patient P2 had a high affinity for antigen, contrary to low-level CD20CD27 B cells, i.e., the canonical surface phenotype of MBCs (Table 1 and Table 2). The same temporally protective increase in specific anti-VZV IgG and without “memory” was observed in the first wave among patients after HSCT. The low CD20+27+ memory B cells may, in part, provide explanation for the phenomena. Furthermore, the clinical model and two peaks of the IgG dynamics observed here after HSCT correspond with the previous finding, with the two phases of the acquisition of B cell memory [30]. Oral mucositis and HSV coexistence are well-described phenomena after HSCT [31,32]: following HSCT anti-HSV is a predictor of ulcerations on oral mucosa [33,34].

In SOT, the memory B lymphocytes (MBCs) and long-lived plasma cells (PCs) are preserved; in HSCT, they are not. These cells are usually depleted (with antithymocyte globulin (ATG) or myeloablative conditioning) and replaced by donor-derived cells with full chimerism [14]. It is noteworthy that, for our patients, the temporary increase in IgG corresponded with a low MBC level and a comparably higher level of active B cells (i.e., CD20+CD23+) (Table 1).

After the first step of B cell differentiation, short-lived plasma cells may be produced; however, their lifespans are generally limited to the course of the infection [30]. The CD20- and CD23-positive B cells that have been demonstrated to be present among our patients are derived from germinal center B-cells with an activatory signature. Interestingly, the subsequent recall responses to HHV antigens during virus replication caused significant increases in the specific IgG level, and a low level of memory B cells; they probably differentiated into long- or short-lived plasma cells [30].

#### 4.4.2. Two Signal and IgG Increase

In vivo viral- and membrane-associated antigens arrive in the lymphatic organ via subcapsular sinuses. The movement and concentration of membrane IgG (regulation of cytoskeleton) are observed after stimulation by corpuscular antigens and B cell responses to soluble or monovalent antigens and require more co-receptors [35]. It is noteworthy that HHVs are the source of a double signal: specific (by surface IgG) and non-specific. Such signals come from pattern recognition receptors, e.g., Toll-Like receptors (TLRs). As described previously for CMV, the TLR signal prompts fast IgG responses without IgM [14]. Furthermore, a study with two signal immunizations (specific and TLR9, i.e., by protein antigens and CpG) resulted in short-lived plasma cell generation and high levels of antibodies that failed in reaching affinity maturation [36] (graphical abstract). This is consistent with low MBC (Table 1) and with the temporary increase in HSV- and VZV-specific IgG in our patients (Figure 1 and Figure 2). Viral genomic DNA of HSV and VZV is a known ligand for TLR9; this has been reported recently [37,38]. This observation corresponds with the latest observation of immunodeficiency with vigorous VZV-susceptibility and TLR-3 variant/mutation [39,40].

Our retrospective analysis has several limitations, such as a small group of patients participating. All previous studies assessing the risk of HSV/VZV reactivation assume an earlier history of the primary infection [1]. This is not clear today. Furthermore, the extrema, i.e., the largest and smallest values of the IgG level, either within a given range (the local or relative extrema in the first wave of immunoreconstitution), or across the entire time (the absolute extrema) is difficult for observation in cohort studies. The model and time-lapse analysis adopted here can be a starting point for further prospective, long-term observations. Specific IgG evolution in a parabolic or hyperbolic manner probably corresponds to various outcomes (for VZV and HSV infection—see Figure 2 and graphical abstract).

There is variability in clinical practices across transplant centers, but preventive antiviral therapy showed low efficacy in our patients; a second wave of the immunoreconstitution of a specific humoral immune response played a significant role (Figure 2 and Figure 3). In contrast, the cumulative incidence of VZV disease after HSCT was higher 2 years after HSCT, with a non-significant influence from conditioning (RIC/MAC) and ATG use [41]. This is generally in line with our observations for VZV (it is noteworthy that P1 and P2 developed VZV reactivation); however, cumulative incidence does not present with differences or timeline, as mentioned above [8]. However, in this cohort study comprising 141 patients, only 6 had VZV disease, and the diagnosis was made without differentiation. The serostatus of the participants is unknown. Therefore, this study was not free from the above-described bias [12], since microbiological, pathological, and/or serological confirmation was not performed [41]. Studies on HSV have not been conducted so far, and our observations show a significant difference in conditioning (RIC + ATG vs. MAC) (Figure 2 and Figure 3).

Furthermore, our study demonstrates only two series of measurements, so there is a high probability of cross-series error. The small sample results from stringent exclusion criteria, put in place to maintain transparency. The primary reason for this limitation is the lack of reproducibility of previous studies, i.e., the practical lack of a simple and accessible laboratory method for verifying the disease due to HHV1–3, contrary to CMV [10]. HSV1/2 and VZV are not viruses that replicate in blood cells (they are, rather, immunity-privileged sites), so the viremia level is not reliable [8], but the Tzank test is rarely used. Similar studies are not continued because heterogeneous groups provide artificial results (no statistical differences) and are contradictory to experimental studies [8]. Moreover, doubts are raised by the poor definition of the disease, the procedure (auto- and allo-HSCT analyzed together), and the laboratory method (semi-quantitative); especially, doubts are raised by the simultaneous use of polyvalent immunoglobulins (containing anti-VZV IgG).

The study highlights the need to clarify definitions and criteria before attempting multicenter and statistical research [9,10,11]. Now, in the COVID-19 era, the use of non-reproducible methods and arbitrary criteria is problematic. Further research with the presented model of specific IgG kinetics and strict disease differentiation are warranted.

## 5. Conclusions

Summing up our observations, a logarithmic/exponential decrease in specific IgG should be considered as a hallmark of the high likelihood of virome dysbiosis and opportunistic HHV1–3 infection. These findings have implications for vaccination and adoptive immunotherapy or passive immunization strategies in patients who have undergone transplants. There is variability in clinical practice across transplant centers, but acyclovir therapy showed low efficacy in our patients until the second wave of immunoreconstitution occurred, with a crucial role for specific IgGs. Furthermore, the cut-off level and threshold of protection against HHV reactivation is higher than that in primary infection [1,2]. Therefore, values of 239 IU/L and 100 RU/mL indicate the point of initiation of active or passive immunization for VZV and HSV, respectively.

Although current EBMT and WMDA standards require the monitoring of several infections (e.g., HIV, HTLV1/2, HBV, HCV, CMV, Treponema), donor immune status against all HHVs and a Tzanck test should be introduced in transplant practices. The lack of differentiation and bias are the main obstacles in statistical HHV1–3 research [12]. Such time-lapse case studies may be the first to build evidence of a pathway and association between immune parameters and HHV diseases. This approach brings translational medicine closer to experimental research, where each patient case is a unique yet strictly and carefully described experiment of nature. In contrast, cohort studies are highly simplified, operating only a few parameters (for example, HCT-CI).

HCT-CI and calculation of the global severity should be extended to include important parameters such as virome homeostasis. HHV1–3 reactivation after immuno-ablative RIC (with ATG) and MAC seems to be just as likely, but the timing of manifestation and outcome are different and may be an introduction to further research in the future.

## Figures and Tables

**Figure 1 biomedicines-12-00566-f001:**
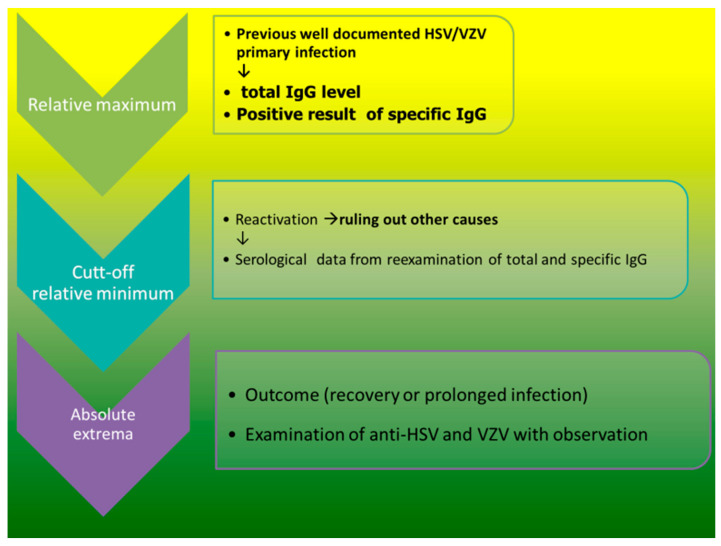
Flow chart of time-lapse observation and data collection. In our analysis, the maxima and minima of IgG level, known collectively as the extrema, are the highest and the lowest values of the specific IgG, respectively; they are either within a given range (e.g., 100 days after HSCT the local or relative extrema) or last the entire time (the absolute extrema).

**Figure 2 biomedicines-12-00566-f002:**
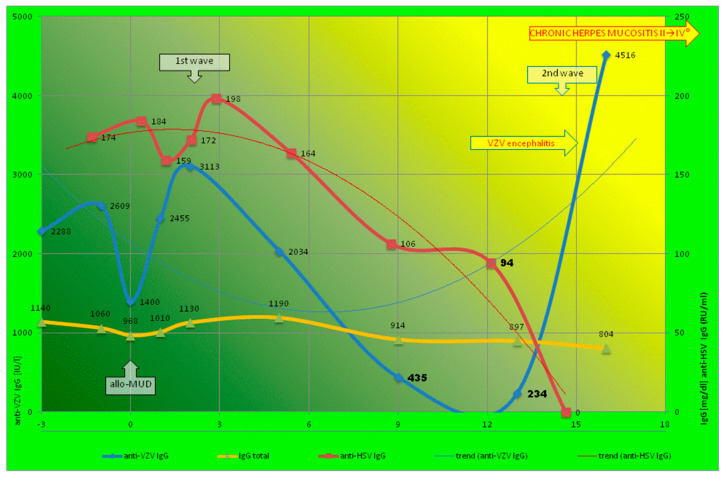
Patient 1 (P1) with ineffective acyclovir therapy and late second-wave immunoreconstitution within a period of 18 months. Levels of specific antibodies were presented in a logarithmic scale. Trend lines (two different directions) reflect different prognoses. Time-lapse measurements are marked with squares and triangles (see figure legends). Crucial dip values (corresponding with HHV reactivation and the onset of the disease) are highlighted in bold.

**Figure 3 biomedicines-12-00566-f003:**
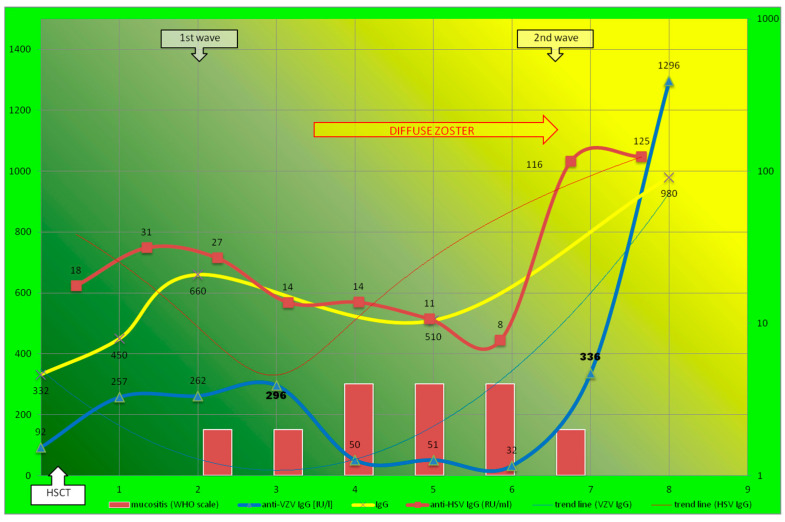
The second patient (P2) after reduced intensity conditioning (RIC) with antithymocyte globulin (ATG) within 9-month follow-up. After HSCT, P2 showed parallel distribution of HSV- and VZV-specific IgG (approximately parallel U-shaped parabolas). Levels of specific antibodies were presented in logarithmic scale (exponentially). Time-lapse measurements are marked with squares and triangles (see figure legends). Crucial dip values (corresponding with HHV reactivation and the onset of the disease) are highlighted in bold.

**Table 1 biomedicines-12-00566-t001:** Clinical and immunological features of patients with HHV reactivation. Their humoral immunodeficiency profiles were determined. B cell and CD4-lymphopenia in patients P1 and P2 could be (in part) caused by cyclosporine A and poor second wave of immunoreconstitution after HSCT (Figure 2 and Figure 3). The memory B cells (CD20- and CD27-positive) were reduced. (The values that are important in pathogenesis are marked with arrows.) Classical complement activation indicated by non-invasive biomarkers of antibody-dependent cytotoxicity (ADCC); these were not observed during VZV reactivation. Cellular compartment of specific immune response (HLA class I and CD8-dependent) was normal.

	P1		P2		
**Gender**	**F**		M		
**Underlying disease**	AML		SAA		
**Conditioning**	Bu-Cy		Flu-Cy-ATG		
**Graft-versus-Host Acute Stage (chronic)**	0 (1)		0 (0)		
**Age**	38		27		
	**initial**	**reactivation**	**initial**	**reactivation**	**NORM**
**Complement C4 (mg/dL)**	15.20	25.2	10.2	15.0	80–60
**β_2_microglobulin (mg/L)**	1.3	3.23	1.8	4.0	1–3
**CsA (ng/mL)**	0	137	0	155	150–250
**WBC (/uL)**	3500	2530	7200	6500	5000–10,000
**%lymphocytes (cells/μL)**					
**CD3**	75.3 (642)	78.1 (↓444)	ND	60.0 (752)	69–72%
**CD4**	↓25.6 (218)	↓↓15.1 (86)	ND	↓25.3 (317)	43–46%
**CD8**	46.1 (393)	64.3 (366)	ND	28.6 (359)	28–30%
**CD20**	12.3 (↓104)	↓↓2.5 (14)	ND	↓5.2 (65)	5–20% (120–600)
**CD20+27+**	↓1.2 (10)	↓↓<0.5% *	ND	↓↓0.7 (9)	1–2% (5–60)
**CD20+23+**	11.2 (96)	1.6 (9)	ND	3.5 (44)	2–4% (10–120)
**CD16+**	8.1 (69)	8(45)	ND	7.7 (97)	NA
**CD16+56+**	7.5 (64)	3.8 (21)	ND	6.7(84)	5–27% (90–590)

* below detection limit.

**Table 2 biomedicines-12-00566-t002:** Evolution of humoral immune parameters and class switch recombination (CSR) for human-herpes-virome-specific antibodies. The stable total class IgG and vaccine-induced anti hepatitis B surface antigen (HBsAg) antibody levels are presented for comparison. It is noteworthy that the IgM antibody response did not proceed four phases of IgG after a foreign antigen challenge. The data are presented as the range of values obtained over the entire observation period (see Figure 1 and Figure 2). Avidity was measured at maximum specific IgG level.

Total and Specific IgG	P1	P2
IgG total (Norm 750–1500 mg/dL)	804–1190	332–980
IgG anti-EBV (Ru/mL)	>200	>200
IgG anti-CMV (RU/mL)	>200	>200
IgG anti-HSV * (RU/mL) (avidity)	178 → 0 (Figure 2) (40 → NA)	7.73 → 124 (Figure 3) (ND)
IgG anti-VZV (IU/L) (avidity )	234 → 4515 (Figure 2) (78)	32 → 1295 (Figure 3) (72)
IgG anti-HbsAg ** (IU/L)	660	160

* It is noteworthy that, for patient P1, a different kinetic was observed, i.e., without a second wave of immunoreconstitution for specific immunity and significant drop of anti-HSV IgG to zero (see Figure 2); avidity could not be determined. ** other serological markers of HBV as well as HIV; HCV were negative. NA—not applicable; ND—not carried out.

## Data Availability

All data generated or analyzed during this study are included in this article.

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
