# Peer review of "Two Waves of Specific B Cell Memory Immunoreconstruction Observed in Anti-HHV1–3 IgG Kinetics after Hematopoietic Stem Cell Transplantation"

_biomedicines, 2024, doi:10.3390/biomedicines12030566_

Round 1
Reviewer 1 Report
Comments and Suggestions for Authors
The submitted manuscript addresses interesting data bridging between basic research and clinical observation regarding humoral factors playing a role in latency and reactivation of herpes viruses in the hosts. The manuscript is well written and presents novel interesting observation. The author introduces a clinical practice and observation approach to assess causes and effect of HHV-1-6 infections in immunocompromised patients after HSCT by collecting a large amount of observation data, which, in my opinion, throws more light on the molecular basis of viral infection. The author studied molecular immune response to herpes viruses during the post-transplant period to define the clinical significance of individual immune response features and HHV reactivation in the post transplant of HSCT patients. The author should focus on some interesting findings on the observation. Furthermore, The author should focus on some interesting findings on the observation of most interesting 2011 Transplant patients in the study. HHV1-?6 are ubiquitous viruses that can establish latency in a large subset of the human population. These are also important pathogens in immunocompromised hosts while latent infection (which may or may not eventuate to disease). Human herpesviruses are difficult to study in immunocompromised hosts due to the continuous antigenic stimulation, in addition to various factors, similar to this immunocompromised patients after HSCT. This article is informative and provides the promoter to recognize factors that may contribute HHV pathogenesis and offer insights into novel strategies to manipulate host immune response to suppress HHV replication as a preventive measures. In your study, you identify interesting findings that may potentially contribute to HHV pathogenesis, as well as guidance for future research on immunoreconstitution. This work has implications to manipulating post-transplant immunity in patients infected with HHV-1-6. There are a few things that are worth addressing in order to improve this study. When I reviewed the article, you were mentioned the major flaws and weaknesses of case analysis. The author should discuss a bit more about the study's design. It is pretty clear from the submission that this is a case analysis design with a small sample size. Due to the often descriptive nature of the results, the study does carry typical threats to validity, such as assessor and confusing bias. The author has taken appropriate precautions to report data and results in a way that minimizes many of these concerns. The use of data from actual cases appeared to be the best approach to obtain a complete picture of HHV-1-3 serostatus, particularly in immunocompromised individuals. Acquisition of data through patient record review eliminates most patient population biases. The information presented provides evidence of the contribution of specific antibodies to the development of HHV-induced diseases and the insufficiency of other immunological responses for establishment of latent infection. Clearly, you have supported the assumption that detection of IgG alone may not be sufficient since patients who showed a decline of IgG levels after HSTC are more likely to develop viral reactivation. The most valuable section of the submission is the adequate breakdown and discussion of the data. The main disadvantage is the relatively small number of patients. This confirms that the author approached the problem pragmatically. During my assessment, the entire analysis may depend upon this small sample. It will be useful to include some information on cost of the intervention. Therefore, unfortunately, the small sample size is a major disadvantage if this study is conducted in a larger cohort. It is important that this study should consider replicating a larger cohort to reflect changes in all proxies to immune protection. Your submission is a nice start to this type of research. Overall, the manuscript presents novel approach to evaluate HHV-1-3 serostatus in immunocompromised individuals after HSCT. The results would be improved by conducting a large cohort studies. The limitation is small case and cross-sectional approach will be significantly increased powers of the study. Thank you again for submitting your observation to our journal.
Author Response
Reviewer 1
- ISSUE: The submitted manuscript addresses interesting data bridging between basic research and clinical observation regarding humoral factors playing a role in latency and reactivation of herpes viruses in the hosts. The manuscript is well written and presents novel interesting observation. The author introduces a clinical practice and observation approach to assess causes and effect of HHV-1-6 infections in immunocompromised patients after HSCT by collecting a large amount of observation data, which, in my opinion, throws more light on the molecular basis of viral infection. The author studied molecular immune response to herpes viruses during the post-transplant period to define the clinical significance of individual immune response features and HHV reactivation in the post transplant of HSCT patients.
The author should focus on some interesting findings on the observation.
ANSWER:
Thanks for Your suggestion.
Content in the manuscript has been modified for highlighting:- Need of standardization of the model and criteria
- the importance of the kinetics of the host↔HHV1-3 interaction (i.e. the relationship between IgG↔disease)
- possibility of preemtive therapy
- Furthermore, The author should focus on some interesting findings on the observation of most interesting 2011 Transplant patients in the study.
Answer:
In accordance with topic, the manuscript has been expanded. In addition to showing bias in cohort studies, attention was paid to what makes our study closer to experimental research. The our “bridge” from bench to bedside is careful selection and meticulous differentiation in transparent (best described) long-therm patient’s history. - HHV1-?6 are ubiquitous viruses that can establish latency in a large subset of the human population. These are also important pathogens in immunocompromised hosts while latent infection (which may or may not eventuate to disease). Human herpesviruses are difficult to study in immunocompromised hosts due to the continuous antigenic stimulation, in addition to various factors, similar to this immunocompromised patients after HSCT. This article is informative and provides the promoter to recognize factors that may contribute HHV pathogenesis and offer insights into novel strategies to manipulate host immune response to suppress HHV replication as a preventive measures. In your study, you identify interesting findings that may potentially contribute to HHV pathogenesis, as well as guidance for future research on immunoreconstitution. This work has implications to manipulating post-transplant immunity in patients infected with HHV-1-6.
There are a few things that are worth addressing in order to improve this study. When I reviewed the article, you were mentioned the major flaws and weaknesses of case analysis. The author should discuss a bit more about the study's design. It is pretty clear from the submission that this is a case analysis design with a small sample size. Due to the often descriptive nature of the results, the study does carry typical threats to validity, such as assessor and confusing bias. The author has taken appropriate precautions to report data and results in a way that minimizes many of these concerns. The use of data from actual cases appeared to be the best approach to obtain a complete picture of HHV-1-3 serostatus, particularly in immunocompromised individuals. Acquisition of data through patient record review eliminates most patient population biases. The information presented provides evidence of the contribution of specific antibodies to the development of HHV-induced diseases and the insufficiency of other immunological responses for establishment of latent infection. Clearly, you have supported the assumption that detection of IgG alone may not be sufficient since patients who showed a decline of IgG levels after HSTC are more likely to develop viral reactivation.
The most valuable section of the submission is the adequate breakdown and discussion of the data. The main disadvantage is the relatively small number of patients. This confirms that the author approached the problem pragmatically. During my assessment, the entire analysis may depend upon this small sample.
ANSWER:- A limitation of our study is the small sample resulting from stringent exclusion criteria. Only after strict selection the transparency of the study is maintained.
- The primary reason for this limitation is the lack of repeatability in past studies, i.e. the practical lack of a simple and accessible laboratory method for verifying the disease due to HHV1-3. The situation is completely different with CMV. However, HSV and VZV are not viruses that replicate in blood cells, so the viral load test is not reliable.
- The Tzank test was used for this purpose. This method shows the actual disease, but it greatly limits the sample. We describe these limitations of our observation at the end of the discussion.
- It will be useful to include some information on cost of the intervention.
ANSWER:- Our study allows for the prediction of reactivation, thus -preemptive therapy. This is important because the main drug, acyclovir, works during the replication period, and long-term use and abuse leads to resistance. (We add sentences in
- The costs of treating a developed disease are much higher and less effective. However, the effectiveness of vaccinations is not tested in a repeatable manner, and the effectiveness criteria vary. We cite last work: Giannelos N et al. doi:10.1080/21645515.2023.2168952.
The authors conclude: Varying incremental cost-effectiveness ratios (ICERs) observed may be associated with different assumptions on the duration of protection, as well as different combinations of structural and disease-related study (model) inputs driving the estimation
Therefore, the effectiveness of vaccinations cannot be precisely estimated if the disease is diagnosed arbitrarily (see answer 3.2 and 3.3.
- Therefore, unfortunately, the small sample size is a major disadvantage if this study is conducted in a larger cohort. It is important that this study should consider replicating a larger cohort to reflect changes in all proxies to immune protection. Your submission is a nice start to this type of research. Overall, the manuscript presents novel approach to evaluate HHV-1-3 serostatus in immunocompromised individuals after HSCT. The results would be improved by conducting a large cohort studies. The limitation is small case and cross-sectional approach will be significantly increased powers of the study. Thank you again for submitting your observation to our journal.
ANSWER- See answer 3.2 and 3.3
- We try to form uniform cohort (see supplementary material and Figure 1) without success.
- Aletrnatively: From these several hundred patients we have serologically examined, data can be collected and statistics can be created. That wouldn't be true. As an example, we cited one research-work where a completely unselected group was studied, the disease was diagnosed arbitrarily, and even questionable methods were used. Therefore it is not surprising that there are no statistical significance. We were looking for something more - temporal and pathophysiological relationships.
Reviewer 2 Report
Comments and Suggestions for Authors
This paper reports HHV1-3 infections in two patients post HSCT on cyclosporine together with a time course of their total and viral specific Ig levels. The patterns show a decline in specific anti-HSV and anti-VZV Ig without a corresponding decline in total Ig in both patients after transplant, then reconstitution of anti VZV in both patients but of anti HSV only in one patient. The anti-VZV levels decline markedly to zero in both patients prior to viral reactivation. The anti-HSV levels declined to zero in one patient with subsequent severe mucositis, whereas the decline was modest in the second patient associated with transient mucositis. The authors emphasize that large cohort studies aggregating multiple parameters draw associations without demonstrating causality, whereas their highly granular analysis of relevant immune parameters in two patients allows for precise timing as a criterion for better assessing the likelihood of causality. They also acknowledge the limitations of a study with just two patients. The authors also point to identifying minimal anti-HHV values prior to expression of disease as providing a useful estimate of protective levels in this context
Comments:
1. The tables should indicate the normal ranges for each parameter as well as the time of the blood draw relative to the time courses shown in the figures.
2. The authors should state more clearly the precedent to this study. i.e. have others measured specific HHV antibody levels before transplant and at the time of clinical disease? The authors extensively cite the vertical transmission/passive antibody literature and cite the CMV literature, but not HSV or VZV.
3. The HSV type is not specified.
4. What is the relative risk of anti-HSV antibody prior to transplant in donor and recipient for development of viral reactivation and disease post transplant?
5. The paper is very hard to read because of poor grammar and English usage.
Comments on the Quality of English Language
Very hard to read due to grammatical, usage errors.
Author Response
Reviewer 2 Comments:
- The tables should indicate the normal ranges for each parameter as well as the time of the blood draw relative to the time courses shown in the figures.
Answer: done - The authors should state more clearly the precedent to this study. i.e. have others measured specific HHV antibody levels before transplant and at the time of clinical disease? The authors extensively cite the vertical transmission/passive antibody literature and cite the CMV literature, but not HSV or VZV.
Answer: done.- Research on infections in HSCT is relatively new, and the introduced indicators do not take into account HHV1-3.
- Literature concerning VZV reactivation in immunodeficiencies and after HSCT is sparse, and in fact CMV predominates. This is due to methodological imperfections (see replies to reviewer 1) but also to inconsistencies in terminology. We cited a seemingly similar work, indicating the source of the bias.
- The HSV type is not specified.
Answer:- Herpes in a person with immunodeficiency or in infants cannot be classified as herpes labialis/genitalis. The treatment is identical. Therefore, most commercial tests and validated methods cover HSV1/2 together.
- Herpes and limited ocular herpes zoster are also indistinguishable (we add the sentence).
- What is the relative risk of anti-HSV antibody prior to transplant in donor and recipient for development of viral reactivation and disease post transplant?
ANSWER:- The limitation of study is fact that we describe two series of results. Therefore statistical analysis was not performed.
See also answer for issue 3 and 5 for reviewer 1
- The limitation of study is fact that we describe two series of results. Therefore statistical analysis was not performed.
- The paper is very hard to read because of poor grammar and English usage.
ANSWER: DONE (the text will be check by native speaker)